# Exploration of N6-Methyladenosine Profiles of mRNAs and the Function of METTL3 in Atherosclerosis

**DOI:** 10.3390/cells11192980

**Published:** 2022-09-24

**Authors:** Yaqing Zhou, Rongli Jiang, Yali Jiang, Yahong Fu, Yerbolat Manafhan, Jinfu Zhu, Enzhi Jia

**Affiliations:** 1Department of Cardiovascular Medicine, The First Affiliated Hospital of Nanjing Medical University, Guangzhou Road 300, Nanjing 210029, China; 2The Friendship Hospital of Ili Kazakh Autonomous Prefecture，Ili & Jiangsu Joint Institute of Health, Yining 835000, China; 3Department of Hypertension, Yili Friendship Hospital, Stalin Road 92, Yining 835000, China; 4Department of Cardiovascular Surgery, the First Affiliated Hospital of Nanjing Medical University, Nanjing 210029, China

**Keywords:** atherosclerosis, smooth muscle cell, RNA-Seq, meRIP-Seq, METTL3

## Abstract

Objectives: N6-methylladenosine (m6A) modification has not been fully studied in atherosclerosis. The objectives of this study were to investigate differentially expressed m6A methylated peaks and mRNAs, along with the regulatory role of methyltransferase 3 (METTL3) in pathological processes of atherosclerosis. Methods: The pathological models of human coronary artery smooth muscle cells (HCASMCs) were induced in vitro. The differentially expressed mRNAs and m6A peaks were identified by RNA-Seq and meRIP-Seq. The potential mechanisms were analyzed via bioinformatic assays. Methylases expression was tested by quantitative real-time polymerase chain reaction (qRT-PCR) and Western blotting (WB) in HCASMCs, and by immunohistochemical assays in 40 human coronary arteries. The knockdown of METTL3 expression in cells was performed by siRNA transfection, and cell proliferation and migration were detected after transfection. Results: We identified 5121 m6A peaks and 883 mRNAs that were expressed differentially in the pathological processes of HCASMCs. Bioinformatic analyses showed that the different m6A peaks were associated with cell growth and cell adhesion, and the 883 genes showed that the extracellular matrix and PI3K/AKT pathway regulate the processes of HCASMCs. Additionally, 10 hub genes and 351 mRNAs with differential methylation and expression levels were found. METTL3 was upregulated in the arteries with atherosclerotic lesions and in the proliferation and migration model of HCASMCs, and pathological processes of HCASMCs could be inhibited by the knockdown of METTL3. The mechanisms behind regulation of migration and proliferation reduced by siMETTL3 are concerned with protein synthesis and energy metabolism. Conclusions: These results revealed a new m6A epigenetic method to regulate the progress of atherosclerosis, which suggest approaches for potential therapeutic interventions that target METTL3 for the prevention and treatment of coronary artery diseases.

## 1. Introduction

Atherosclerosis is a gradual pathological process consisting of plaque formation in the arterial walls, which is the leading cause of many cardiovascular diseases [1,2]. In the progress of atherosclerosis, different kinds of cells are involved [3]. Among these kinds of cells, smooth muscle cells (SMCs) remain in a slow proliferative state and stable phenotype before the occurrence of atherosclerosis [4]. During the development of atherosclerosis, the abnormal proliferation and migration of SMCs are among the most important pathological processes [5]. Hence, it is significant to determine the regulatory networks and mechanisms behind the interaction of proliferation and migration of SMCs in atherosclerosis.

Epigenetic modification—including histone modification, RNA methylation, and DNA methylation—has become a hot topic recently, having been observed in most tumors and many other diseases [6,7]. Previous studies have reported numerous types of RNA modifications [8]. Among them, N6-methylladenosine (m6A) modification is considered to be the most common RNA modification [9]. M6A modification has been confirmed to be involved in a diversity of conditions, including cardiovascular disease [10], tumorigenesis [11], and nonalcoholic fatty liver disease [12]. However, the functions and effects of m6A modification in the proliferation and migration of SMCs in atherosclerosis remain unknown, calling for further studies.

The process of m6A modification is adjusted and controlled by many methylases, including “writers”, “readers”, and “erasers” [13]. Methyltransferase 3 (METTL3) is a member of the writers, which was found to catalyze m6A modification in RNAs [14]. METTL3 is important in various phases of RNA life, including alternative splicing, mRNA decay, translation, and nuclear transport [15,16,17]. In cellular processes, METTL3 was confirmed to be critical for promoting the proliferation and migration of many cancer cells [18]. However, whether METTL3 could impact the proliferation and migration of SMCs in atherosclerosis has not been fully elucidated.

Here, we conducted meRIP and RNA sequencing analyses to look for m6A peaks and mRNAs that are differentially expressed in pathological processes of human coronary artery smooth muscle cells (HCASMCs). Then, the expression of methylases was validated via molecular biology experiments. The knockdown of METTL3 expression was performed by siRNA transfection, and cell proliferation and migration were detected after transfection. This work may open a new direction for the treatment of atherosclerosis from the perspective of HCASMC dysfunction in upcoming years.

## 2. Materials and Methods

### 2.1. Preparation and Pathological Analysis of Coronary Arteries

The inclusion, preparation, and pathological analysis of 40 human coronary artery samples were described in our previous studies [19]. The Ethics Committees of Nanjing Medical University and the First Affiliated Hospital of Nanjing Medical University have approved the ethical considerations and experimental programs.

### 2.2. Cell Cultures

HCASMCs were obtained from the biotechnology company Sigma (Sigma, St. Louis, MO, USA). HCASMCs were cultivated in an incubator that was maintained at 37 °C and 5% CO_2_ using DMEM (Gibco, Waltham, MA, USA) with 10% fetal bovine serum (FBS, Gibco, Waltham, MA, USA) and 1% double-resistant solution (penicillin and streptomycin, Beyotime, Shanghai, China). The pathological model of HCASMCs was cultured with oxidized low-density lipoprotein (ox-LDL) (Yiyuan Biotechnologies, Guangzhou, China) at a concentration of 50 mg/L for 24 h to build proliferation and migration models with an abnormal lipid environment, which were established and verified in our previous study [20].

### 2.3. RNA Preparation

Total RNA of HCASMCs was isolated with RNA-easy isolation reagent (Vazyme, Nanjing, China) as recommended by the manufacturer, followed by reducing the ribosomal RNA content with the Ribo-Zero rRNA Removal Kit (Illumina, San Diego, CA, USA). Next, we measured the concentration of total RNA using a NanoDrop 2000 (Thermo Fisher, Waltham, MA, USA) and used OD260/OD280 as the purity index. When the value of OD260/OD280 was higher than 1.8 and lower than 2.1, the RNA purity was identified as eligible, and the RNA could be used for subsequent processing steps.

### 2.4. RNA-Seq Library Construction and Sequencing

Cloud-Seq Biotech (Shanghai, China) was entrusted to perform the RNA sequencing. The NEBNext^®^ Ultra™ II Directional RNA Library Prep Kit (New England Biolabs, Ipswich, MA, USA) was applied to build the RNA libraries. To control the quality of the libraries, the Bioanalyzer 2100 system (Agilent Technologies, Inc., Santa Clara, CA, USA) was used. The paired-end sequencing of the library was performed using a HiSeq 4000 instrument (Illumina, San Diego, CA, USA).

### 2.5. MeRIP-Seq Library Construction and Sequencing

The MeRIP-Seq service was provided by Cloud-Seq Biotech as well. Briefly, the GenSeq™ m6A RNA IP Kit (GenSeq Inc., Shanghai, China) was applied to perform the m6A RNA immunoprecipitation against its specifications. Both the input sample and the immunoprecipitation samples were used for RNA-Seq library generation using the NEBNext^®^ Ultra II Directional RNA Library Prep Kit (New England Biolabs, Ipswich, MA, USA). The library quality was evaluated using the Bioanalyzer 2100 system as mentioned previously, and library sequencing was performed as previously described.

### 2.6. Bioinformatics Analysis

Kyoto Encyclopedia of Genes and Genomes (KEGG) as well as Gene Ontology (GO) analyses were performed using the clusterprofiler package in R software (version 4.1.0, Auckland, New Zealand). We predicted the protein–protein interaction (PPI) network of the selected genes via the Search Tool for the Retrieval of Interacting Genes (STRING) database (https://string-db.org/, 28 February 2022). Genes with a score of ≥0.15 were chosen, and the results were presented using Cytoscape (Version 3.8.2, New York, NY, USA). The genes with the top five maximal clique centrality (MCC) values calculated by CytoHubba were considered to be hub genes in the process of proliferation and migration of HCASMCs.

### 2.7. Quantitative Real-Time Polymerase Chain Reaction (qRT-PCR)

Total RNA was extracted as described above, and cDNA was prepared using TransScript One-Step gDNA Removal (Vazyme, Nanjing, China). cDNA was then amplified using a StepOnePlus (Applied Biosystems, Waltham, MA, USA) instrument with SYBR qPCR Master Mix (Vazyme, Nanjing, China). The oligonucleotide sequences of the primers are shown in Table 1. Glyceraldehyde-3-phosphate dehydrogenase (GAPDH) was used as an internal reference gene to estimate gene expression. Relative gene expression was calculated using the 2^−ΔΔct^ method.

### 2.8. Western Blotting (WB)

HCASMCs were cultured in 6-well plates at a density of 5 × 10 ^5^/well. Then, the RIPA Lysis Kit (Beyotime, China) mixed with phosphatase inhibitor (Beyotime, China), protease inhibitor (Beyotime, China), and phenylmethanesulfonyl fluoride (Beyotime, China) at 50× was applied to lyse the HCASMCs. The same amount of total protein was separated on SDS–PAGE gel by electrophoresis and then transferred onto a polyvinylidene fluoride membrane (Millipore, Billerica, MA, USA). Membranes were blocked with blocking buffer (Beyotime, Shanghai, China) at room temperature, and then incubated overnight at 4 ℃ with primary antibodies toMETTL3 (1:1000, Proteintech, Wuhan, China). On the next day, the membranes were incubated in secondary antibodies for 1 h at room temperature. Proteins were imaged using the Tanon-5200 automatic gel imaging system (Tanon, Shanghai, China).

### 2.9. Immunohistochemistry of Coronary Artery Tissues

We performed immunohistochemistry experiments on 40 human coronary specimens with different stages of atherosclerotic lesions, as described in our previous article [19]. The sections of the human coronary tissues were incubated with primary antibodies to METTL3 (1:400; Proteintech, Wuhan, China). Then, the sections were washed and incubated with secondary antibodies (7076s, Cell Signaling, Woburn, MA, USA) for 1 h. The average of integrated optical density (IOD) of five random regions of each sample analyzed using ImageJ software was used for statistical analysis in the next steps.

### 2.10. Cell Counting Kit 8 (CCK-8)

To detect the proliferation capacity of HCASMCs, the CCK-8 assay kit (APExBIO, Houston, TX, USA) was applied. A total of 5000 HCASMCs per well in 100 μL of medium was inoculated in 96-well plates (Corning, New York, NY, USA). The absorbance at 450 nm was measured at the same time every day with a microplate reader (Multiskan FC, Thermo Fisher, Waltham, MA, USA).

### 2.11. Transwell Assay

First, 200 μL of medium with 20,000 cells was plated into the upper chambers of transwell chambers (Corning, New York, NY, USA). Then, 600 μL of DMEM containing 15% FBS was added to the lower chambers. After 24 h, the cells in the upper chamber were removed, and the cells on the lower side of the filter membrane were fixed in 4% glutaraldehyde. Finally, crystal violet was used to stain the fixed cells, which were counted and photographed under a microscope.

### 2.12. Statistical Analysis

Statistical analyses were performed using SPSS (version 25.0, Chicago, IL, USA), GraphPad Prism (version 8.0, San Diego, CA, USA), and R (version 4.1.0, Auckland, New Zealand). The Shapiro–Wilk normality test was used to test the normality of the data. We used unpaired or paired *t*-tests when the data were parametric, while unpaired Mann–Whitney U tests were used when the data were nonparametric. Data are presented as the mean ± SD unless otherwise stated. When the *p*-value was lower than 0.05, the result was statistically significant.

## 3. Results

### 3.1. Overview of mRNA m6A Methylation in Proliferation and Migration Models of HCASMCs

To establish proliferation and migration model of HCASMCs with an abnormal lipid environment, cells were cultured with ox-LDL for a concentration of 50 mg/L for 24 h. We calculated the methylation peaks in the models of HCASMCs and the control group. Methylation peaks in the model group were less than those in the control group. The results showed that 20,957 m6A peaks in the control group and 16,825 m6A peaks in the model group were detected (Figure 1A). There were 1234 hypermethylated peaks and 3887 hypomethylated peaks identified in mRNAs, and the volcano diagrams depict the differentially expressed m6A peaks (Figure 1B). The top 20 differentially expressed peaks are presented in Table 2. Additionally, we used CIRCOS software to analyze the distribution of mRNA methylation peaks on the chromosomes (Figure 1C). The results showed that the distribution of the methylation peaks of mRNA on each chromosome was different between the two groups. The hypermethylated m6A peaks were found in all other chromosomes except for chrY (Figure 1D).

### 3.2. Analysis of Sources of mRNA Methylation and Motif Analysis

Then, we studied each m6A peak in order to evaluate the peaks systematically. As a result, 90% of m6A peaks in the control group and 88% of m6A peaks in the model group were within genic regions. In the control group, 73% of genic peaks were localized near the stop codon and coding sequence (CDS), while 10% were found in the 5’ untranslated regions (5’UTRs) and 3’ untranslated regions (3ʹUTRs). In the model group, 12% were found in the 5ʹUTRs and 3ʹUTRs, and 69% of genic peaks were localized near the stop codon and CDS (Figure 2A). M6A peaks were significantly correlated with two different coordinates: immediately following near the end of the 5ʹUTRs and the start of the CDS, and near the end of the CDS and the beginning of the 3ʹUTRs (Figure 2B). DREME software was used to scan the sequence of methylated peaks of each group of samples to find meaningful motif sequences. The top three motifs of methylated peaks in the model group are shown in Figure 2C. The top three motifs were HGGARGA, GCDGCDG, and GAAGAWV, with *p*-values of 1.8 × 10^−130^, 6.1 × 10^−58^, and 1.0 × 10^−52^, respectively (H = A/C/U, R = A/G, D = A/G/U, W = A/U, V = A/C/G).

### 3.3. GO and KEGG Enrichment Analyses of m6A Sites

GO analyses were performed (Figure 3A). In total, 1114 biological process (BP) terms, 113 cellular component (CC) terms, and 110 molecular function (MF) terms were enriched in different peaks. The most enriched BP term was cell growth (*p* = 1.16 × 10^−8^). Cell–cell junction was the most enriched CC term (*p* = 4.71 × 10^−9^). The most enriched MF term was GTPase regulator activity (*p* = 1.78 × 10^−11^). Next, a KEGG pathway enrichment analysis was performed to analyze the most significantly enriched pathways for different peaks (Figure 3B). A total of 59 pathways was enriched in different peaks, and the top five enriched pathways were Alzheimer’s disease (*p* = 0.007), Huntington’s disease (*p* = 0.006), coronavirus disease (COVID-19) (*p* = 3.96 × 10^−6^), endocytosis (*p* = 0.005), and Rap1 signaling pathway (*p* < 0.001).

### 3.4. Characteristics of mRNA Expression Profiles in Pathological Models of HCASMCs

Next, we calculated the number of coding genes in the pathological models of HCASMCs and the control group. The results showed that 8424 mRNAs in the control group and 7473 mRNAs in the model group were found (Figure 4A). With the filtering criteria of FC ≥ 1.5 and *p* < 0.05, 883 differentially expressed genes—including 172 upregulated genes and 711 downregulated genes—were identified (Figure 4B). In addition, the top 20 highly expressed genes and top 20 lowly expressed genes are presented as a heatmap in Figure 4C.

### 3.5. GO and KEGG Analyses of Differentially Expressed Genes

In total, 495 BP terms, 21 CC terms, and 64 MF terms were enriched in differentially expressed genes (Figure 5A). The most enriched BP terms were extracellular matrix organization (*p* = 2.57 × 10^−5^), extracellular structure organization (*p* = 2.66 × 10^−5^), and external encapsulating structure organization (*p* = 2.83 × 10^−5^). Collagen-containing extracellular matrix (*p* = 2.05 × 10^−4^), basal plasma membrane (*p* = 5.27×10^−3^), and basal part of cell (*p* = 8.31 × 10^−3^) were the most enriched CC terms. The most enriched MF terms were growth factor binding (*p* = 1.92 × 10^−4^), extracellular matrix binding (*p* = 1.50 × 10^−4^), and fibroblast growth factor binding (*p* = 3.22 × 10^−4^). KEGG pathway enrichment analysis was applied to analyze the most significantly enriched pathways for different peaks (Figure 5B). In total, 21 signaling pathways were enriched in differentially expressed genes, and the top five enriched pathways were the PI3K-Akt signaling pathway (*p* = 0.03), coronavirus disease (COVID-19) (*p* = 0.006), MAPK signaling pathway (*p* = 0.026), human papillomavirus infection (*p* = 0.049), and IL-17 signaling pathway (*p* = 3.93 × 10^−4^).

### 3.6. Identification of Hub Genes and Conjoint Analysis of RNA-Seq and MeRIP-Seq

The PPI network among the 883 differentially expressed genes was established (Figure 6A), and the 10 hub genes selected using the MCC algorithm are presented in Figure 6B. The 10 hub genes included IL6 (MCC = 1.60 × 10^7^), IL1β (MCC = 1.60 × 10^7^), MMP3 (MCC = 1.58 × 10^7^), MMP1 (MCC = 1.53 × 10^7^), IGFBP3 (MCC = 1.52 × 10^7^), LCN2 (MCC = 1.40 × 10^7^), COL1A1 (MCC = 1.38 × 10^7^), GDF15 (MCC = 1.37 × 10^7^), TGFβI (MCC = 1.28 × 10^7^), and SMAD3 (MCC = 1.06 × 10^7^). Genes were sorted from high to low by their MCC values, which was found to be the most accurate of all the methods available in the CytoHubba. The specific information of ten hub genes from RNA-Seq and meRIP-Seq is shown in Appendix A. Based on the above findings, an intersection analysis was conducted to select the mRNAs with differential levels of both methylation and expression in the pathological models of HCASMCs. Thus, the differentially expressed m6A methylated peaks and mRNAs were involved, and 351 mRNAs with both differential methylation and expression levels were found. Among them, 30 mRNAs were highly expressed and hypermethylated, while 15 mRNAs were highly expressed and hypomethylated. In addition, 30 mRNAs were downregulated and hypermethylated, while 276 mRNAs were downregulated and hypomethylated (Figure 6C). The MCC value of IL6 was the highest among the genes, while the adjusted *p*-value of expression between the experimental group and control group was higher than 0.05 (adjust *p* = 0.37). Moreover, m6A methylation peaks were not found in IL6. Therefore, we speculate that IL1β was the key factor.

### 3.7. Validation of the Expression of Methylases in HCASMCs and the Expression of METTL3 in Coronary Specimens

The relative expression of 13 methylases between the pathological model of HCASMCs and the control groups was detected. Among the writers, METTL3, METTL14, and RBM15B were upregulated, while KIAA1429 was downregulated (Figure 7A). The two erasers, including FTO and ALKBH5, were found to be upregulated (Figure 7B). Four readers—YTHDC1, HNRNPC, YTHDF2, and YTHDF3—were detected to be upregulated (Figure 7C). Considering the general importance of METTL3 in the proliferation and migration of different kinds of cells, we further detected protein levels of METTL3 in HCASMCs, and the results showed the high expression level of METTL3 in the pathological model of HCASMCs (Figure 7D). To further validate the expression of METTL3 in atherosclerosis, immunohistochemical analysis of METTL3 was performed (Figure 8). There was a statistically significant difference between Stage 1 and Stage 2 (*p* < 0.05). In Stage 3, the expression of METTL3 was upregulated, and there was a significant difference (*p* < 0.001). In the advanced stages of atherosclerosis, METTL3 expression was significantly upregulated between Stage 4 and Stage 1 (*p* < 0.001). In Figure 8F, the bar chart of the IOD value is presented.

### 3.8. Regulatory Role of METTL3 in Proliferation and Migration Models of HCASMCs

To explore the regulatory role of METTL3 in proliferation and migration models of HCASMCs, we designed three siRNAs to knock down the expression of METTL3. The protein level of METTL3 was detected after transfection of siRNAs (Figure 9A). After knocking METTL3 down, the expression of the proliferation-associated protein, PCNA, was tested to determine whether the proliferation ability of the HCASMCs changed. The results showed that the knockdown of METTL3 could inhibit the proliferation of HCASMCs (Figure 9B). To confirm this result, CCK-8 analysis was performed, and the results were consistent (Figure 9C). In addition, the migration of HCASMCs was inhibited by the knockdown of METTL3, as detected by transwell assay (Figure 9D). Considering the function and upregulation of METTL3, 1234 upregulated m6A peaks on 1096 mRNAs, which were hypermethylated, may be targets of METTL3. To predict the potential mechanisms, GO and KEGG analyses were performed with 1096 mRNAs. The results are shown in Appendix A. The most enriched BP term was cellular protein complex disassembly (*p* = 1.71 × 10^−18^) (Appendix A). Intracellular was the most enriched CC term (*p* = 7.15 × 10^−17^) (Appendix A). The most enriched MF term was binding (*p* = 5.07 × 10^−7^) (Appendix A). Next, a KEGG pathway enrichment analysis was performed (Appendix A). The top five enriched pathways were Ribosome (*p* = 8.26 × 10^−15^), Huntington’s disease (*p* = 8.91 × 10^−7^), Alzheimer’s disease (*p* = 3.51 × 10^−6^), Oxidative phosphorylation (*p* = 5.24 × 10^−5^), and Parkinson’s disease (*p* = 4.85 × 10^−4^). As a whole, the mechanisms behind the regulation of migration and proliferation reduced by knockdown of METTL3 are concerned with protein synthesis and energy metabolism.

## 4. Discussion

Atherosclerosis remains a potential cause of cardiovascular and cerebrovascular diseases, which can lead to myocardial infarction and stroke, which constitute the main causes of death around the world [21]. The abnormal proliferation and migration of SMCs play a major role in the pathophysiology of atherosclerosis [22]. The aim of this study was to provide a comprehensive survey of m6A methylation and expression features of mRNAs in the proliferation and migration of HCASMCs, and to elucidate the regulatory role of METTL3 to provide treatment directions and possibilities for atherosclerosis. In this research, we identified 5121 m6A peaks and 883 mRNAs that were differentially expressed in the proliferation and migration of HCASMCs by meRIP-Seq and RNA-Seq. GO and KEGG enrichment analyses showed that the different m6A peaks were associated with cell growth and cell adhesion. GO and KEGG enrichment analyses of the differentially expressed genes showed that the processes of the extracellular matrix and PI3K/AKT pathway regulate the pathological processes of HCASMCs. In total, 10 hub genes and 351 mRNAs with differential levels of both methylation and expression were found, and proliferation and migration of HCASMCs could be inhibited by knockdown of METTL3.

In 2012, Dan Dominissini and Kate D. Meyer proposed m6A methylation sequencing as a new method to map m6A peaks at the transcriptome level and verified more than 12,000 methylation sites on human mRNAs [23,24]. So far, the widespread application of next-generation sequencing (NGS) in epigenetic research has contributed to evaluating the distribution of m6A sites more accurately, promoting a comprehensive understanding of the function of m6A modification [25]. The transcriptome-wide m6A methylome has been applied to study many diseases, such as high myopia [26], age-related cataracts [27], gastric cancer [28], etc. In the field of atherosclerosis, Deng et al. researched the m6A levels of RNA from peripheral blood mononuclear cells in cases of coronary artery disease [29]. However, the m6A levels of smooth muscle cells that are involved in the processes of atherosclerosis have not been fully explored. In our study, we conducted meRIP sequencing and RNA sequencing analyses on proliferation and migration models of HCASMCs. We found that m6A was enriched in the CDS and 3’UTR regions in smooth muscle cells, as found in mammals. The m6A methylation peak in the experimental group was found to be higher at the beginning of the CDS and decreased at the end of the CDS. According to our results, we speculate that when smooth muscle cells enter the proliferative and migratory states, the changes in genetic and microenvironmental factors would be most concentrated at the beginning and end of the CDS. In addition, large numbers of m6A modifications at the CDS or 3‘UTR sites may regulate the stability, transport, and translation of RNA [30]. Further research is required in order to verify these findings.

Based on the bioinformatics analyses of meRIP-Seq and RNA-Seq, we found that different m6A peaks and differentially expressed mRNAs were associated with cell growth, cell adhesion, the extracellular matrix, and the PI3K/AKT pathway. The extracellular matrix is a complex network composed of various macromolecules, including collagen, non-collagen, elastin, proteoglycan, and aminoglycan, which is dynamic to maintain tissue homeostasis. An abnormal extracellular matrix affects the progress of diseases by directly promoting the proliferation, migration, and differentiation of cells [31]. Therefore, the extracellular matrix around smooth muscle cells is highly correlated with the pathological process of smooth muscle, providing a research direction for regulating the occurrence and development of atherosclerosis. In addition, the PI3K/AKT pathway has been confirmed to be one of the most important cellular signaling pathways, playing a crucial role in the regulation of cell growth, migration, and proliferation [32]. In our previous study, we also verified that the PI3K/AKT/mTOR pathway was activated in the proliferation and migration models of HCASMCs [33]. In the future, we could focus on the use of inhibitors of the PI3K/AKT pathway to inhibit the proliferation and migration of smooth muscle cells in order to slow the development of atherosclerosis.

The writers of methylases mainly consist of METTL3/14, KIAA1429, WTAP, ZC3H13, and RBM15/15B [34,35,36]. A previous study found that the overexpression of METTL3 in hepatocellular carcinoma (HCC) may promote the degradation of the SOCS2 gene, resulting in an abnormal JAK/STAT signaling pathway, causing the proliferation and migration of HCC cells [37]. In patients with HCC, overexpression of METTL3 can reduce the expression of RDM1 in an m6A-dependent manner, promoting the survival, proliferation, and stability of hepatocellular carcinoma cells [38]. In addition, METTL3 may regulate the expression of USP7 through modification of m6A methylation and may promote the invasion of hepatocellular carcinoma cells [39]. Next, we further searched for articles about studies on smooth muscle cells. Qin et al. found that METTL3 mRNA and protein were found upregulated in the proliferation and migration of smooth muscle cells and hypoxic rat models [40]. When the expression of METTL3 was silenced, proliferation and migration of pulmonary artery smooth muscle cells were inhibited. In this study, we found that the expression of METTL3 in the model of HCASMCs was higher than that in the control group. In coronary segments, the upregulation of METTL3 confirmed its role in atherosclerosis. The darkly stained regions were located in the intima and media of the arteries, suggesting the important regulatory function of METTL3 in HCASMCs. By knocking down its expression via siRNA of METTL3, we found that it had an inhibitory effect on the proliferation and migration of smooth muscle cells, proving that METTL3 has an important regulatory effect on the proliferation and migration of coronary artery smooth muscle cells, which was the same as in previous articles. Through conjoint analysis of meRIP-Seq and RNA-Seq, 60 mRNAs were found to be hypermethylated and differentially expressed, representing possible targets of METTL3, which merit further experiments in the future.

There are some limitations to this study. Firstly, the cellular model of HCASMCs is only limited to 24 h post treatment, and some molecules and m6A modifications that require longer stimulation and higher concentrations to be expressed differentially were not included, which is one of the limitations to the study. Secondly, the molecular mechanisms and functional experiments in this study were less than perfect, and the specific mechanisms by which METTL3 may participate in the pathological process of HCASMCs and the development of atherosclerosis were not explored. Thus, more comprehensive and in-depth research is needed. Thirdly, carefully designed experiments on animal models of atherosclerosis, along with clinical trials, will be needed in order to further validate our theoretical predictions. To increase the potential clinical utility of the identified mRNAs, the functional roles of the mRNAs identified in the present study should be clarified and validated in further studies.

In conclusion, 5121 m6A peaks and 883 mRNAs were differentially expressed in the proliferation and migration of HCASMCs, which were enriched in cell growth, cell adhesion, the extracellular matrix, and the PI3K/AKT pathway. The m6A peaks were enriched at the CDS and 3’UTR regions, and there were differences between the control group and the experimental group at the beginning and the end of the CDS. In addition, METTL3 promotes proliferation and migration of HCASMCs, and knockdown of METTL3 could inhibit this process in HCASMCs. This study’s findings suggest that the methylase METTL3 and mRNAs with differential levels of both methylation and expression may be targets for regulating HCASMC proliferation and migration in atherosclerosis, which could provide a new diagnostic basis and therapeutic target for cerebrovascular and cardiovascular diseases.

## Figures and Tables

**Figure 1 cells-11-02980-f001:**
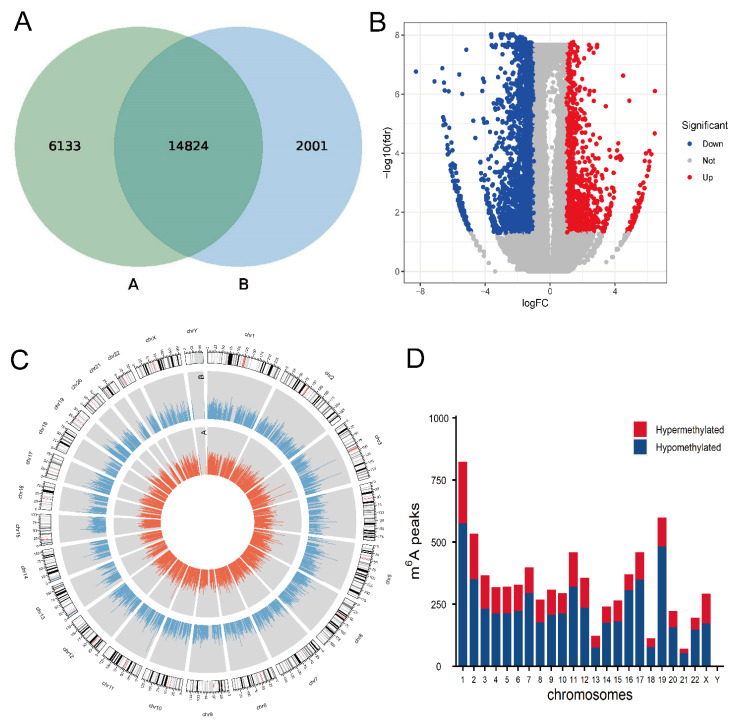
Overview of mRNA m6A methylation map in proliferation and migration models of HCASMCs: (**A**) The number of methylation sites of the control and experimental groups. (**B**) The volcano plots of the differentially m6A peaks. (**C**) Visualization of m6A at the chromosome level in the control and experimental groups. (**D**) Distribution of differentially methylated m6A sites with significance in chromosomes.

**Figure 2 cells-11-02980-f002:**
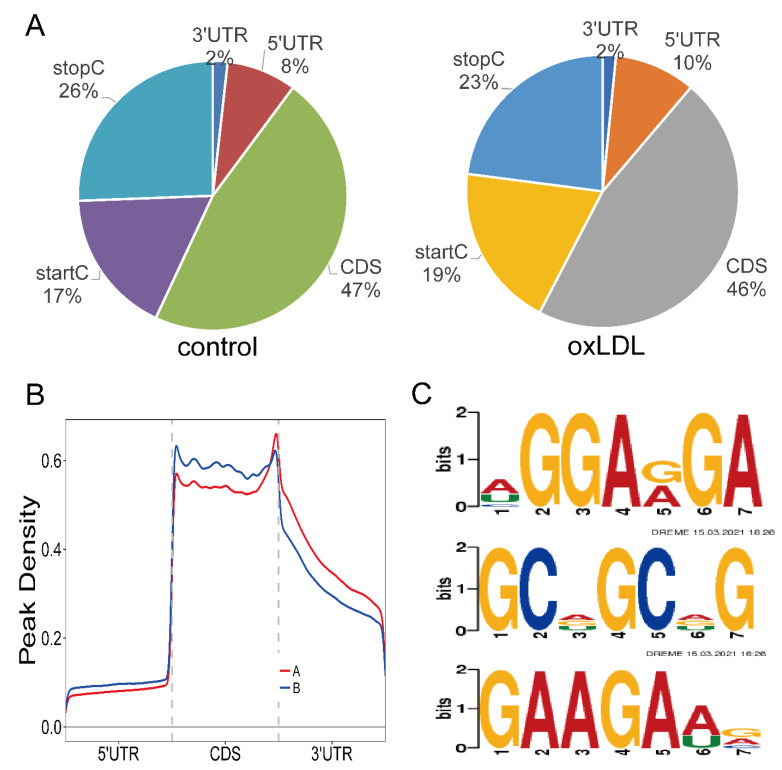
Analysis of sources of mRNA methylation and motif analysis: (**A**) Pie charts showing m6A peaks’ distribution in different gene contexts. (**B**) Preferential location of m6A in mRNA. Each transcript is divided into three parts, including a 5’ untranslated region, coding DNA sequence, and 3’ untranslated region. (**C**) The top three motifs enriched across m6A peaks identified from the experimental group.

**Figure 3 cells-11-02980-f003:**
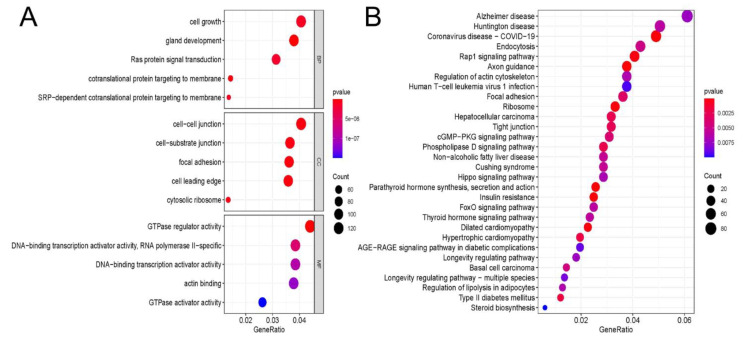
GO and KEGG analyses of m6A peaks: (**A**) Major Gene Ontology terms were significantly enriched for the hypermethylated and hypomethylated genes. (**B**) The top 30 significantly enriched pathways for the hypermethylated and hypomethylated genes.

**Figure 4 cells-11-02980-f004:**
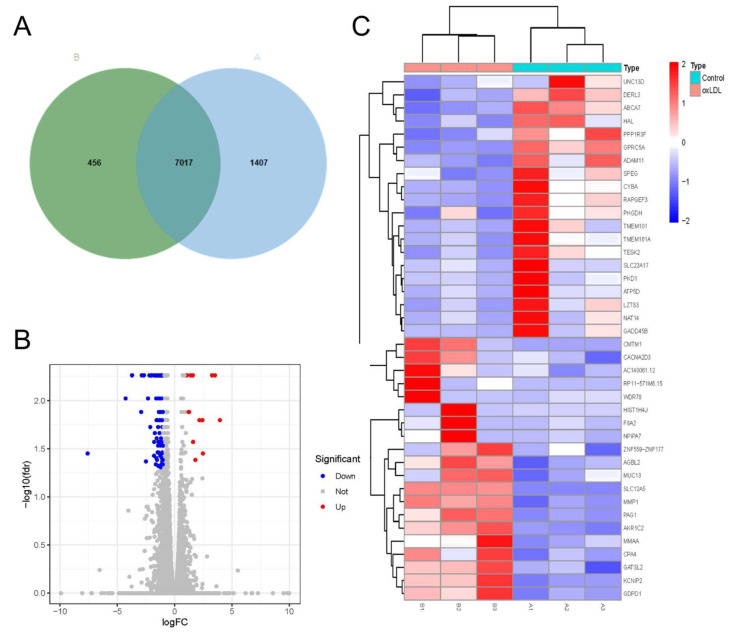
Characteristics of mRNA expression profiles in proliferation and migration models of HCASMCs: (**A**) The number of coding genes in the proliferation and migration models of HCASMCs and the control group. (**B**) The volcano plots of the differentially expressed genes. (**C**) The heatmap of the top 20 highly expressed genes and top 20 lowly expressed genes.

**Figure 5 cells-11-02980-f005:**
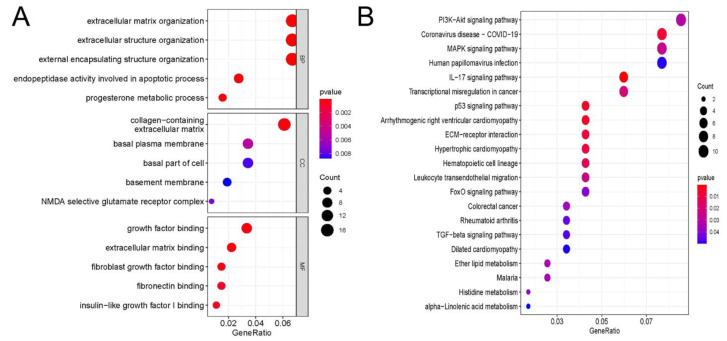
GO and KEGG analyses of differentially expressed genes: (**A**) Major Gene Ontology terms were significantly enriched for the differentially expressed genes. (**B**) The top 20 significantly enriched pathways for the differentially expressed genes.

**Figure 6 cells-11-02980-f006:**
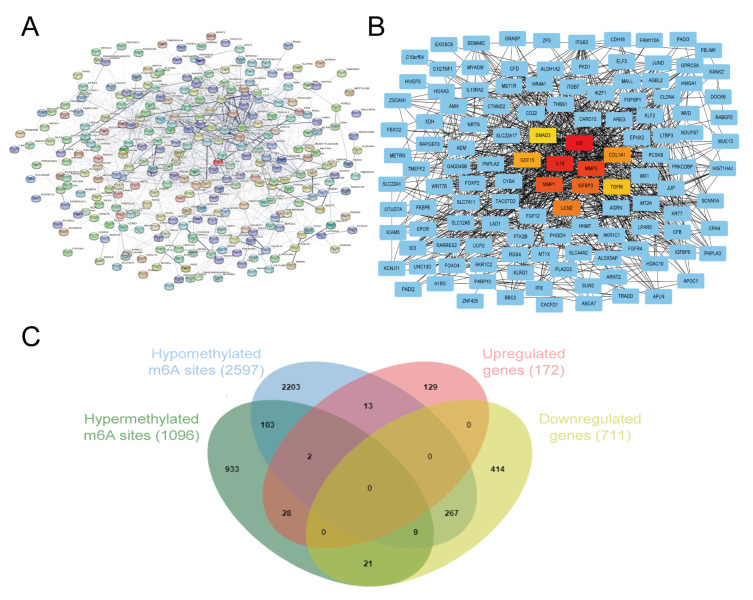
Identification of hub genes and conjoint analysis of RNA-Seq and MeRIP-Seq: (**A**) The PPI network of the differentially expressed genes. (**B**) The 10 hub genes calculated by cytoHubba. (**C**) The intersection analysis of RNA-Seq and MeRIP-Seq.

**Figure 7 cells-11-02980-f007:**
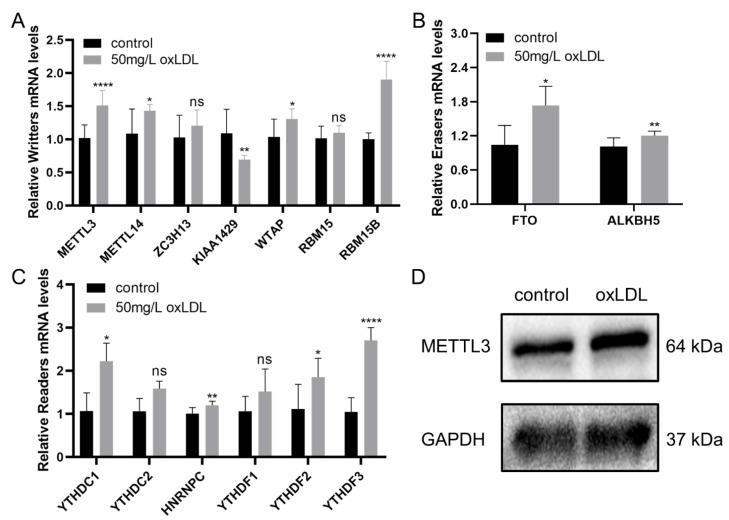
The expression of methylases in HCASMCs: (**A**) The expression of writer methylases by qRT-PCR. (**B**) The expression of eraser methylases by qRT-PCR. (**C**) The expression of reader methylases by qRT-PCR. (**D**) The expression of METTL3 by WB. * *p* < 0.05, ** *p* < 0.01, **** *p* < 0.0001.

**Figure 8 cells-11-02980-f008:**
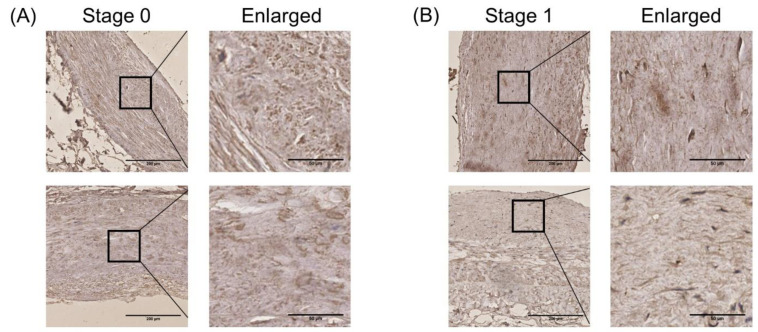
The expression of METTL3 at different stages of atherosclerosis, analyzed by immunohistochemical assay: (**A**) The expression level of METTL3 at Stage 0 of atherosclerosis. (**B**) The expression level of METTL3 at Stage 1 of atherosclerosis. (**C**) The expression level of METTL3 at Stage 2 of atherosclerosis. (**D**) The expression level of METTL3 at Stage 3 of atherosclerosis. (**E**) The expression level of METTL3 at Stage 4 of atherosclerosis. (**F**) The bar chart and differences between the other groups and Stage 1. Stage 0, normal tunica intima; Stage 1, fatty streak tunica intima; Stage 2, fibrous plaques tunica intima; Stage 3, atherosclerotic tunica intima; Stage 4, secondary affection tunica intima. Low magnification (40×); scale bar: 200 μm. High magnification (200×); scale bar: 50 μm. * *p* < 0.05. *** *p* < 0.001.

**Figure 9 cells-11-02980-f009:**
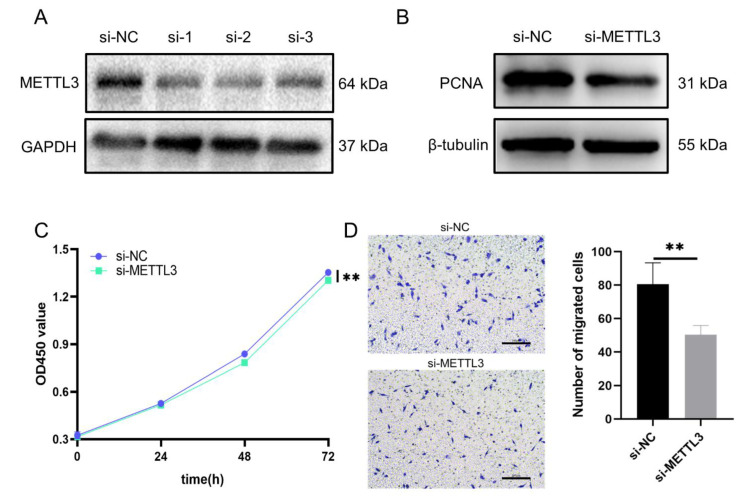
Regulatory role of METTL3 in proliferation and migration models of HCASMCs: (**A**) The protein levels of METTL3 were detected after transfection of siRNAs. (**B**) Knockdown of METTL3 could downregulate the proliferation-related protein PCNA. (**C**) CCK-8 analysis showed that the knockdown of METTL3 could inhibit the proliferation of HCASMCs. (**D**) The migration of HCASMCs was inhibited by knockdown of METTL3, as detected by transwell assay. ** *p* < 0.01.

**Table 1 cells-11-02980-t001:** Summary of the oligonucleotide primer and siRNA sequences.

Gene	Forward Primer	Reverse Primer
GAPDH	GTCTCCTCTGACTTCAACAGCG	ACCACCCTGTTGCTGTAGCCAA
METTL3	CTATCTCCTGGCACTCGCAAGA	GCTTGAACCGTGCAACCACATC
METTL14	CTGAAAGTGCCGACAGCATTGG	CTCTCCTTCATCCAGATACTTACG
ZC3H13	CGGACAGTGATGCCTACAACAG	TCTGTGAGGTGCGAGGGACTAA
KIAA1429	CTCTTCCTAACCACAGTGAACC	AGCCTTTCTATTTCCCCTTCAC
WTAP	GCAACAACAGCAGGAGTCTGCA	CTGCTGGACTTGCTTGAGGTAC
RBM15	GTTGTGGCTTATGTGGAGTTTAC	CACTTAAAACACCGGCATTGG
RBM15B	TGGTAACCTGGACCACAGCGTA	GGTTCTGGAACTTGAGGAAGGC
YTHDC1	TCAGGAGTTCGCCGAGATGTGT	AGGATGGTGTGGAGGTTGTTCC
YTHDC2	GAAAGCTCCTGAACCTCCACCA	GGTTCTACTGGCAAGTCAGCCA
HNRNPC	CCAGCAACGTTACCAACAAGACA	CCTCCACATCAGATTTCTTGACC
YTHDF1	CAAGCACACAACCTCCATCTTCG	GTAAGAAACTGGTTCGCCCTCAT
YTHDF2	TAGCCAGCTACAAGCACACCAC	CAACCGTTGCTGCAGTCTGTGT
YTHDF3	GTTCCTCAGCTCTTTTCTCCAG	TGGATCAAGGCCATATTTTCAAAG
FTO	GGTATCTCGCATCCTCATTGG	GAGGAAGGTCTCACAAGCAG
ALKBH5	CCAGCTATGCTTCAGATCGCCT	GGTTCTCTTCCTTGTCCATCTCC
si-METTL3-1	GCUGCACUUCAGACGAAUUTT	AAUUCGUCUGAAGUGCAGCTT
si-METTL3-2	GCUCAACAUACCCGUACUATT	UAGUACGGGUAUGUUGAGCCT
si-METTL3-3	GCAAGAAUUCUGUGACUAUTT	AUAGUCACAGAAUUCUUGCAC

**Table 2 cells-11-02980-t002:** The top 20 differentially methylated m6A peaks.

Regulation	Gene	Fold Change	Chromosome	Start	End	Peak Length	*p*-Value
Up	CTB-133G6.1	87.9	Chr19	7445841	7445880	39	8.11 × 10^–9^
	PSMF1	87.2	Chr20	1093905	1093920	15	3.80 × 10^−7^
	TCAF2	75.6	Chr7	143400621	143400710	89	2.58 × 10^−6^
	SLA2	69.6	Chr20	35242781	35242840	59	5.05 × 10^−6^
	SEPT9	69.2	Chr17	75447561	75447610	49	1.90 × 10^−6^
	OR2T8	68.5	Chr1	248084361	248084560	199	7.86 × 10^−6^
	THAP6	66.7	Chr4	76474814	76474930	116	1.04 × 10^−5^
	TCAF2	62.5	Chr7	143318541	143318596	55	1.81 × 10^−5^
	TMF1	59.7	Chr3	69068977	69069160	183	2.38 × 10^−6^
	ZNF774	56.6	Chr15	90908721	90908940	219	4.74 × 10^−6^
Down	DHRS3	308.2	Chr1	12632755	12632840	85	1.36 × 10^−9^
	ARSI	139.8	Chr5	149677321	149677860	539	3.40 × 10^−9^
	MDGA1	99.7	Chr6	37665261	37665766	505	9.99 × 10^−10^
	TULP2	96.6	Chr19	49398254	49398419	165	8.49 × 10^−8^
	XYLT1	95.6	Chr16	17202541	17202874	333	1.77 × 10^−7^
	CLEC18A	93	Chr16	69993641	69993749	108	3.82 × 10^−9^
	AL953854.2-002	92.9	Chr9	65661850	65662160	310	1.11 × 10^−7^
	HYDIN	90.2	Chr16	70969854	70970017	163	7.61 × 10^−9^
	MRGPRG	87.2	Chr11	3239601	3239980	379	1.65 × 10^−7^
	C10orf54	83.3	Chr10	73512731	73512759	28	5.26 × 10^−7^

## Data Availability

All data and materials have been made available.

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
