# Peer review of "Exploration of N6-Methyladenosine Profiles of mRNAs and the Function of METTL3 in Atherosclerosis"

_cells, 2022, doi:10.3390/cells11192980_

Round 1

Reviewer 1 Report

In the study, titled "N6-Methyladenosine and expression features of mRNAs and role of METTL3 in dysfunction of smooth muscle cells in atherosclerosis", Zhou Y et al., describe the role of m6-methylation in regulation of proliferation and migration of human coronary artery smooth muscle cells (HCASMCs). Authors show how knockdown of METTL3 in HCASMCs reduces proliferation and migration. Overall, the hypothesis is interesting but some of the results are confusing and require more clarity.

1. Title needs to be rephrased to show the main finding of the study.

2. Abstract does not describe the rationale for the study or novelty of the findings.

3. The results regarding proliferation and migration in figure 9 after METTL3s silencing are confusing. Large body of literature shows how methylation is related to ablation of proliferation and synonymous to differentiation. In contrast, authors find that blocking METTL3 reduces PCNA and proliferation of HCASMCs. This needs to be further assessed or the authors need to discuss this point in the discussion section.

4. Similarly, both migration and proliferation are reduced by siMETTL3 and mechanism behind regulation of these independent biological/cellular processes is not clear.

5. Author should at least mention once the in vitro cell model for HCASMCs in the results section.

6. Also, authors should add a limitation of the study in the discussion documenting how the cellular model is only limited to 24hrs post treatment.

7. Can the authors explain why they see reduced methylation peaks in figure 1 in the model group compared to control group yet the expression of methylation writers are increased?

Reviewer 2 Report

This article reports SMCs play an important part in the pathophysiology of atherosclerosis. This manuscript provided a comprehensive survey of m6A methylation and expression features of mRNAs. They suggest that METTL3 and the mRNAs with both differential methylation was the key factor to induced atherosclerosis.

Specific comments

Methods:

1. In the part of “Preparation and pathological analysis of coronary arteries”. The source of the specimen seems to be only one race, is there any difference between different races?

Results:

1.     This article suggests that ten hub genes and 351 mRNAs with both differential methylation and expression levels were found. Please try to conclude which one was the key factor and explain why?

2.In figure7 data A (qRT-PCR data showing the expression of methylase) suggested that RBM15B is also specific. Why was it not discussed?

Round 2

Reviewer 1 Report

There are no further comments